# Exploring the Mediating Role of Parental Anxiety in the Link between Children’s Mental Health and Glycemic Control in Type 1 Diabetes

**DOI:** 10.3390/ijerph20196849

**Published:** 2023-09-27

**Authors:** Evija Silina, Maris Taube, Maksims Zolovs

**Affiliations:** 1Department of Psychiatry and Narcology, Riga Stradins University, LV-1007 Riga, Latvia; maris.taube@rsu.lv; 2Statistical Unit, Riga Stradins University, LV-1007 Riga, Latvia; maksims.zolovs@rsu.lv; 3Institute of Life Sciences and Technology, Daugavpils University, LV-5401 Daugavpils, Latvia

**Keywords:** adolescent, type 1 diabetes, depression, anxiety, glycated hemoglobin, parental anxiety

## Abstract

Pediatric diabetes type 1 diabetes mellitus (T1D), as a chronic, incurable disease, is associated with psychoemotional and socioeconomic burden for the whole family. Disease outcomes are determined by the metabolic compensation of diabetes, characterized by the level of glycated hemoglobin (HbA1c). The caregivers play a critical role in the metabolic control of children with T1D. The aim of this study was to investigate which environmental factors may explain the relationship between diabetes compensation and anxiety and depression of a child. The cross-sectional interdisciplinary study recruited dyads from adolescents and their parents (N = 251). Adolescent and parent groups were screened for anxiety and depression. General linear model (GLM) mediation analysis was performed to determine the potential mediating effect of surrounding factors on the relationship between dependent variables (glycated hemoglobin) and independent variables (the child’s anxiety and depression symptoms). The study revealed that the relationship between HbA1c and the child’s anxiety and depression is fully mediated (B = 0.366, z = 4.31, *p* < 0.001) by parental anxiety. Diabetes metabolic control in adolescents with T1D is related to adolescents’ mental health via parents’ anxiety. This means that parents’ anxiety plays a more significant role in the level of HbA1c than the anxiety and depression of the adolescent.

## 1. Introduction

Type 1 diabetes (T1D) is the most common autoimmune disorder in childhood [1]. Three-quarters of all cases of type 1 diabetes are diagnosed in individuals at less than 18 years of age. Over 1.2 million children and adolescents have type 1 diabetes worldwide, and the global incidence of T1D has increased annually by 3% since the 1980s. Over half (54%) are under 15 years of age [2].

Patients with diabetes mellitus are forced to completely change their lifestyle to follow a specific regime which includes regular blood glucose level control, injecting insulin multiple times a day, while considering carbohydrate intake and physical activity. The only treatment is lifelong insulin administration, which is often a psychological trauma for both the patient and his or her relatives and may cause the development of mental disorders such as anxiety and depression, as evidenced by previous studies. Symptoms of anxiety and depression and the corresponding disorders are the most frequently reported emotional problems and mental comorbidities in youth with T1D [3]. A systematic literature review by Benton et al. (2023) found a high prevalence of mental disorders and associated needs among people with type 1 diabetes [4].

Anxiety and depression symptoms negatively affect glycemic control [3,5,6,7,8,9,10]. A correlation was observed between anxiety and depression level and glycemic control, as well as three-way interaction among glycated hemoglobin (HbA1c)—a key criterion for diabetes compensation—the frequency of blood glucose monitoring, and diabetes-related stress [11]. Statistically significant three-way interaction indicates that one or more of the three possible two-way interactions (HbA1c—frequency of blood glucose monitoring; HbA1c—diabetes-related stress; and the frequency of blood glucose monitoring and diabetes-related stress) differ across the levels of a third variable. It means that the interaction between glycated hemoglobin and the frequency of glycemia monitoring may vary by level of diabetes-related stress. On the other hand, the relationship between HbA1c and diabetes-related stress can also be affected by the frequency of blood glucose checking. These variables interact because the relationship between an dependent (HbA1c) and independent variable (the frequency of blood glucose monitoring or diabetes-related stress) changes depending on the value of a third variable (the frequency of blood glucose monitoring or diabetes-related stress). There is an inverse relationship between self-control and psychosocial complications. The scientific literature indicates that adolescents with T1D are 2.3 times more likely to suffer from anxiety and depression than their healthy peers [5].

The exact cause of the disease is unknown, but there is evidence to suggest that both genetic and environmental factors (with the proposed agents including viral infections, immunizations, diet, vitamin D deficiency and perinatal factors) found in variable combinations in individual patients are involved in the development of T1D [12,13]. The incidence of T1D in childhood has increased, and the age at diagnosis has decreased due to environmental changes during the last half of the twentieth century [12].

Chronic somatic diseases are significant risk factors for the development of anxiety and depression. The complex interrelationship between depressive disorders and chronic disease has important implications for both chronic disease management and the treatment of depression [14]. Various studies have been conducted in children with chronic somatic diseases. Knight et al. found that children with type 1 diabetes had higher levels of depression and suicidal ideation than their peers with systemic lupus erythematosus/mixed connective tissue disease, and it was associated with HbA1c [15]. According to parents’ reports, patients with diabetes having history of acute hypo—or hyperglycemia and higher glycated hemoglobin as well as patients with poorly controlled asthma symptoms manifest more problems with self-regulation than their healthy peers but less than children with attention deficit hyperactivity disorder [16].

In this study, we investigated which environmental factors may explain the relationship between the glycated hemoglobin level and anxiety and depression of the child. The investigated factors were divided into the following groups (Figure 1): individual factors (inner child environment) and surrounding factors (parental and socio-economic environment). Since parents are responsible for creating the surrounding environment and controlling the interaction of the child with this environment, we expected there to be an indirect effect of the mental health conditions (anxiety and depression) of the child on the level of glycated hemoglobin in blood via the environment created by the parents (parents’ mental health conditions, the self-assessment of health status, life satisfaction, marital status, education, employment, income, relationships in family, the number of children in the family and persons in the household).

## 2. Materials and Methods

### 2.1. Participants

This cross-sectional interdisciplinary study recruited adolescents aged 12–18 of both genders with T1D (N = 251) and their parents (N = 251). One of the child’s parents, mostly mothers, participated in the survey. Data from 251 parents (86.9% female, Mage = 46.33; 13.1% male, Mage = 44.06), age range = 30–58 years) were screened. The research took place in the Children’s Clinical University Hospital, the Outpatient Department of Endocrinology and in GP (family doctor) practices. The participants were selected according to availability during their routine visit to the outpatient departments. Parents and their adolescents were informed about the purpose and methods of the research. Permission was obtained from Riga Stradins University Ethics Committee (No. 6-1/07/46) to conduct the study.

The research group (N = 502; 251 child–parent dyads) consisted of adolescents with T1D (N = 251) and their parents (N = 251) (Table 1).

### 2.2. Measurements

The mental health conditions of the respondents were assessed by determining the symptoms of anxiety and depression. Two tools were used: The Generalized Anxiety Disorder Scale-7 (GAD-7) and the Patient Health Questionnaire 9 (PHQ-9) Scale. The GAD-7 is a 7-item self-report scale recommended for screening generalized anxiety disorder and evaluating its severity. Participants were asked the degree to which seven symptoms (e.g., feeling nervous, anxious or on edge) had bothered them during the preceding 2 weeks. The severity of the anxiety was assessed in the range of 0–21 depending on the answers [17]. The PHQ-9 is a 9-item self-report measure of depression severity, validated by Kroenke et al., 2001 (adaptation and standardization in Latvia [18]). The questionnaire asks how often participants have been bothered by certain problems in the past 2 weeks. Then, nine symptoms were listed [19]. As a severity measure, the PHQ-9 score can range from 0 to 27, since each of the 9 items can be scored from 0 (not at all) to 3 (nearly every day) [20]. The socio-demographics and clinical data of the study groups were obtained using a questionnaire developed by the authors (Appendix A) and from additional information available in the patients’ medical records. The glycemic control of patients was assessed using the last glycated hemoglobin (HbA1c) values from patients’ medical records. The formation of glycated hemoglobin is a non-enzymatic process that occurs throughout the life cycle of red blood cells (approximately 120 days) by attaching plasma glucose to the hemoglobin molecule. The value of glycated hemoglobin is determined by the concentration of glucose in the blood during the last 3–4 months (average life span of an erythrocyte). HbA1c shows the degree of diabetes compensation. HbA1c was determined using a high-performance liquid chromatographic (HPLC) method [21]. A target range of HbA1c for all age-groups of < 7.5% is recommended [22].

### 2.3. Statistical Data Analysis

The first exploratory analysis comprised a forward stepwise linear regression model to identify possible predictors of the child’s glycated hemoglobin level among individual factors (inner child environment) and surrounding factors (parental and socio-economic environment), (see the table in the Appendix A). At each step, variables were chosen based on *p*-values (*p* < 0.05) and model fit assessed via R^2^ and AIC. To evaluate the validity of the results, the following assumptions of linear regression were tested: (1) the assumption of normal residual distribution was assessed by the inspection of the normal Q-Q plots of residuals. The Durbin–Watson test was used to test the assumption of independence of observations (autocorrelation). The residual plot was used to test the assumption of homoskedasticity. To assess the assumption of multicollinearity, the variance inflation factors (VIFs) were calculated. The presence of multivariate outliers was tested according to Cook [23].

To measure the strength of association between glycated hemoglobin (HbA1c) and GAD-7 and PHQ-9, Spearman’s correlation was used because data were non-normally distributed, as assessed using Shapiro–Wilk test and Q-Q plots. Semi-partial Spearman’s correlation was conducted to measure the strength of linear association between glycated hemoglobin (HbA1c) and the child’s GAD-7 and PHQ-9, holding the parent’s GAD-7 and PHQ-9 constant for the child’s GAD-7 and PHQ-9.

General linear model (GLM) mediation analysis was performed to determine the potential mediating effect on the relationship between dependent and independent variables. To test the significance of the mediated effect (ME) for non-normally distributed data, bootstrapping procedures (10,000 bootstrapped samples) were used. The 95% CI was calculated by determining the unstandardized MEs at 2.5th and 97.5th percentiles. All statistical analyses were performed by using the Jamovi 2.3.28 statistical software (https://www.jamovi.org; accessed on 26 September 2023.) and R (v.4.1.2; https://www.r-project.org; accessed on 26 September 2023). An alpha level of 0.05 was used for all the statistical analyses.

## 3. Results

A first exploratory analysis (a forward stepwise regression) showed that data better fit the following regression model: (F (5, 245) = 49.9, *p* < 0.001, adj. R^2^ = 0.49). This indicated that approximately 49% of the variance in the child’s glycated hemoglobin is explainable by the parent’s Generalized Anxiety Disorder Scale-7 (GAD-7) score and the parent’s education (Table 2), where the parent’s GAD-7 score showed a more trusting effect than education level (Figure 2). We also tried to build a regression model, where we first included the child’s GAD-7 and the Patient Health Questionnaire 9 (PHQ-9) scale, and then the parent’s GAD-7 and PHQ-9 score, which resulted in an insignificant *p*-value of the child’s GAD-7 and PHQ-9 score, as well as markedly lower R^2^ than the final regression model.

Therefore, a series of correlations were calculated that showed the strong positive association between the child’s glycated hemoglobin and the child’s GAD-7 (r_s_ = 0.682, n = 251, *p* < 0.001) and PHQ-9 (r_s_ = 0.656, n = 251, *p* < 0.001) (Figure 3), as well as the strong positive correlation between the child’s glycated hemoglobin and the parent’s GAD-7 (r_s_ = 0.751, n = 251, *p* < 0.001) and PHQ-9 (r_s_ = 0.710, n = 251, *p* < 0.001). Then, semi-partial correlation was conducted. After controlling the parent’s GAD-7 and PHQ-9 scores, the association between the child’s glycated hemoglobin and the child’s GAD-7 and PHQ-9 scores became insignificant (*p* > 0.05).

GLM mediation analysis was performed to assess the mediating role of the parent’s GAD-7 and PHQ-9 on the linkage between the dependent variable (HbA1c) and independent variables (child’s GAD-7 and PHQ-9 and parent’s education). The final GLM mediation model is shown in Figure 4. The results (Table 3) revealed that the total effect of the child’s GAD-7 (B = 0.479, z = 4.30, *p* < 0.001) on HbA1c was significant, but the total effect of the child’s PHQ-9 (B = 0.166, z = 1.49, *p* = 0.135) was not significant. With the inclusion of the mediating variable (parent’s GAD-7), the impact of the child’s GAD-7 on HbA1c was found to be insignificant (B = 0.113, z = 0.98, *p* = 0.326), and the impact of the child’s PHQ-9 on HbA1c was found to also be insignificant (B = 0.068, z = 0.74, *p* = 0.458). The indirect effect of the child’s GAD-7 on HbA1c though the parent’s GAD-7 was found to be significant (B = 0.366, z = 4.31, *p* < 0.001), and the indirect effect of the child’s PHQ-9 on HbA1c though the parent’s GAD-7 was found to also be significant (B = 0.098, z = 2.56, *p* = 0.010). This indicates that the relationship between the dependent variable (HbA1c) and independent variables (child’s GAD-7 and child’s PHQ-9) is fully mediated by the parent’s GAD-7 [24].

## 4. Discussion

The results of this study contribute to the existing literature by highlighting the mediating role of parental anxiety in the relationship between a child’s anxiety and depression and glycated hemoglobin (HbA1c) levels. This suggests that the impact of a child’s mental health conditions on diabetes management is not solely direct but also influenced by the mental health of the parents.

The mediation analysis performed in this study reveals a novel insight—the mediating role of parental anxiety in the relationship between a child’s anxiety and depression and HbA1c levels. This suggests that a parent’s psychological state can significantly influence a child’s diabetes management outcomes. While previous studies have examined individual effects of both the child’s and the parent’s mental health on diabetes management [10,11,25,26], this study brings to light a possible pathway through which parental anxiety might indirectly impact a child’s glycemic control. This finding emphasizes the need to involve parents in diabetes care and address their mental health concerns as part of a comprehensive approach to managing the condition in children.

Data analysis did not confirm a statistically reliable relationship between the child’s individual environment and HbA1c, as well as socioeconomic factors in the surrounding environment. Examining parental factors by mediation analysis we found that the relationship between HbA1c and the child’s General Anxiety Disorder 7 score (GAD-7) and the child’s Patient Health Questionnaire 9 score (PHQ-9) is fully mediated by parental anxiety. No statistically significant association with other parental factors was found.

The effects of the parent’s and the child’s anxiety and depression on HbA1c have been investigated in many previous studies. The integrative literature review by Rechenberg et al. (2017), including data from 20 studies, confirms the association of anxiety with higher HbA1c levels and more frequent depressive symptoms in youth with type 1 diabetes (T1D) [10]. The literature review by Bass et al. (2020) finds that parental stress predicts a worsening in the control of HbA1c levels, while parental diabetes-specific distress predicts an increase in children’s depressive symptoms [26]. Similar results are found in the meta-analyses by Buchberger et al. and Akbarizadeh et al. [11,27]. Our study confirms the previous ones by showing a strong positive correlation between anxiety and depression and HbA1c in both groups of respondents—parent and child.

However, not all studies confirm a positive correlation between mental health conditions and HbA1c. The Stallwood cohort study (2005) cited in the systematic review by Tsiouli et al., contrary to other studies, shows lower HbA1c levels with higher parental stress [28].

Separate mediation studies have been conducted in diabetology. The data are insufficient to comprehensively judge the possibilities of improving the care of children with T1D and to pay timely attention to risk factors affecting the quality of life. Most mediation studies analyze the association of various factors with the metabolic compensation of diabetes. Some mediation studies are summarized in Table 4.

Factors characterizing mental state (mood [7,15], stress [9], sleep [29]), socioeconomic factors (income [30]), psychological factors (relationships) and diabetes monitoring [31] were used as independent variables. The dependent variable in all of the studies was HbA1c. Mediators are key factors that require more research to improve the metabolic compensation of diabetes. It is necessary to work on reducing diabetes-specific stress and burden, improving self-care to reduce the HbA1c level, thereby avoiding long-term complications caused by the disease. In our study, similarly to Vesco et al.’s study, the child’s anxiety was defined as the independent variable. Vesco et al. study finds diabetes distress (the emotional response to the burdens of living with diabetes) as a mediator between the child’s anxiety and HbA1c [9]. Since anxiety is considered a component of diabetes distress, our study can be compared to that by Vesco et al. In the mentioned study, double mediation was analyzed through automatic negative thinking and diabetes distress. The indirect effect of anxiety on HbA1c through diabetes distress was significantly greater than the indirect effect through both automatic negative thinking and diabetes distress. However, Vesco et al.’s study did not analyze parents’ mental conditions, so it is not possible to fully compare the results. None of the studies reviewed found parental anxiety as a mediator between the child’s anxiety and HbA1c.

Up to now, no interdisciplinary studies have been conducted in Latvia to determine the prevalence of depression in the adolescent population suffering from type 1 diabetes mellitus. Similarly, the influence of parental psychological status on the metabolic control of T1D has not been studied in Latvia either. Research experience from other countries indicates the need to study adolescents with T1D and their caregivers in Latvia as well.

It is important to note some limitations of the study. Being a cross-sectional study, it could establish causality, and the direction of the relationships observed remained uncertain. Additionally, the study was based on self-reported measures for anxiety and depression, which might have introduced bias. Moreover, the study focused on a specific population of adolescents with type 1 diabetes and their parents, so the findings might not be directly applicable to other age groups or diabetes types. Despite the limitations, our study has important practical and clinical implications. For families with an adolescent with T1D, making parental anxiety reduction a primary target could improve diabetes compensation.

## 5. Conclusions

In conclusion, this study provides insights into the complex relationship between mental health conditions (anxiety and depression) and glycated hemoglobin levels in children with type 1 diabetes. Parents are the first and most important providers of comprehensive diabetes care. Their priority is to teach their child to take responsibility for controlling the disease. This is best if the mental state of the caregivers is compensated. The study highlights the role of parental anxiety as a mediator in this relationship, suggesting that parental mental health plays a significant role in diabetes management outcomes for children. This reinforces the importance of addressing mental health concerns in both children and their parents as part of diabetes care strategies. As demonstrated by the mediation studies analyzed in the discussion section, diabetes is not only an endocrinological disease. It also touches areas such as psychology, psychiatry, socioeconomical environment and family functioning in general. This determines what was suggested in previous studies—diabetes care should be multidisciplinary, including not only endocrinologists and primary care doctors, but also psychologists, psychiatrics, social workers, and other specialists. It is important to note that working with parents, including preventive measures to improve mental health, is a mandatory condition of interdisciplinary diabetes care.

## Figures and Tables

**Figure 1 ijerph-20-06849-f001:**
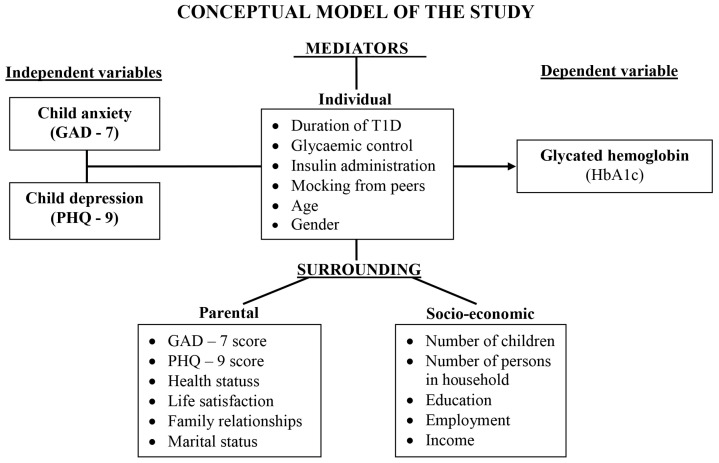
Explanatory factors used in mediation analysis. The conceptual model of the research.

**Figure 2 ijerph-20-06849-f002:**
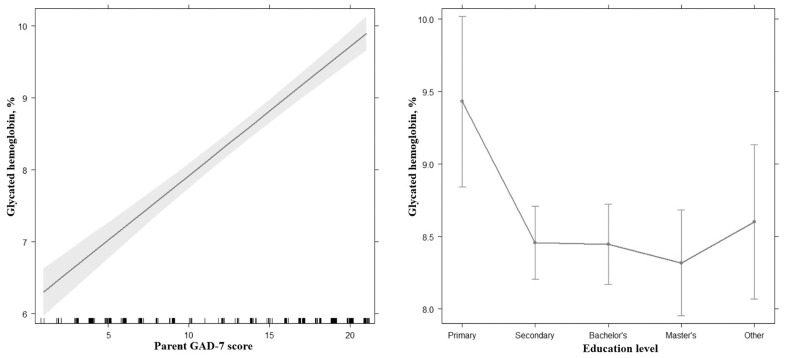
Effect plots of final regression model showing relationship between dependent variable (glycated hemoglobin) and independent variables (parent’s GAD-7 score and parent’s education level).

**Figure 3 ijerph-20-06849-f003:**
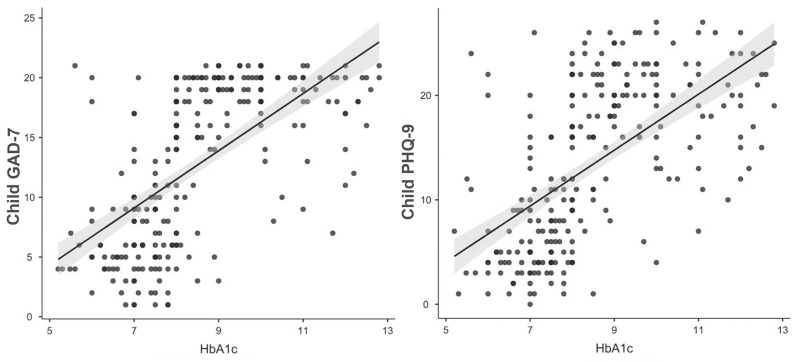
Association between child’s glycated hemoglobin and child’s GAD-7 and PHQ-9.

**Figure 4 ijerph-20-06849-f004:**
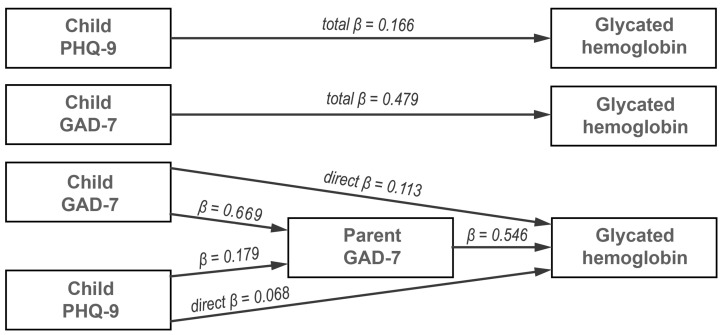
GLM mediation analysis, which included glycated hemoglobin as the dependent variable, the parent’s GAD-7 score as the mediator variable, and the child’s GAD-7 and PHQ-9 score as independent variables.

**Table 1 ijerph-20-06849-t001:** Distribution of research participants and applied methods.

Study Population (N = 502; 251 Child–Parent Dyads)	
Adolescents with T1D (N = 251)	Parents (N = 251)
Methods	
Generalized Anxiety Disorder 7-item (GAD-7) Scale ↓ Symptoms of anxiety	Patient Health Questionnaire 9 (PHQ-9) Scale ↓ Symptoms of depression

**Table 2 ijerph-20-06849-t002:** Final regression model of relationship between child’s glycated hemoglobin and the environment created by the parents.

Predictor	Estimate	95% CI	t	*p*
Intercept	7.04	6.35–7.73	20.12	<0.001
Parent’s GAD-7 score	0.18	0.16–0.20	14.86	<0.001
Parental education:				
Secondary—Primary	−0.97	0.33–1.61	−3.00	0.003
Bachelor’s—Primary	−0.98	0.33–1.64	−2.97	0.003
Master’s—Primary	−1.11	0.42–1.81	−3.16	0.002
Other—Primary	−0.83	0.04–1.62	−2.07	0.040

**Table 3 ijerph-20-06849-t003:** GLM mediation analysis, which included the glycated hemoglobin as the dependent variable, the parent’s GAD-7 score as the mediator variable, and the child’s GAD-7 and PHQ-9 score as independent variables. Confidence intervals were calculated by using the bootstrap procedure (10,000 bootstrapped samples). The reported betas are complete standardized effect sizes.

Type	Effect	Estimate	95% CI of Estimate		β	z	*p*
			Lower	Upper			
Indirect	Child’s GAD-7 → Parent’s GAD-7 → Glycated hemoglobin	0.096	0.052	0.139	0.366	4.319	<0.001
	Child’s PHQ-9 → Parent’s GAD-7 → Glycated hemoglobin	0.022	0.004	0.038	0.098	2.565	0.010
Component	Child’s GAD-7 → Parent’s GAD-7	0.665	0.519	0.815	0.669	8.828	<0.001
	Parent’s GAD-7 → Glycated hemoglobin	0.144	0.093	0.194	0.546	5.612	<0.001
	Child’s PHQ-9 → Parent’s GAD-7	0.151	0.035	0.263	0.179	2.607	0.009
Direct	Child’s GAD-7 → Glycated hemoglobin	0.029	−0.029	0.089	0.113	0.982	0.326
	Child’s PHQ-9 → Glycated hemoglobin	0.015	−0.025	0.055	0.068	0.742	0.458
Total	Child’s GAD-7 → Glycated hemoglobin	0.126	0.068	0.183	0.479	4.300	<0.001
	Child’s PHQ-9 → Glycated hemoglobin	0.037	−0.011	0.086	0.166	1.496	0.135

**Table 4 ijerph-20-06849-t004:** Mediation studies in diabetology.

Study Authors (Year)	Study Population	Independent Variable	Dependent Variable	Mediator
Vesco et al. (2021) [9]	Adolescents with T1D, N = 264	Anxiety	HbA1c	Diabetes-specific distress
Frye et al. (2019) [29]	10–16-year-old children, N = 111	Sleep duration		Diabetes management
Hagger et al. (2018) [7]	Adolescents with T1D, N = 450	Depressive symptoms		Diabetes-specific distress
Thomas et al. (2018) [30]	Youths and their parents, N = 390	Family income		Parenting constructs
Hilliard et al. (2013) [31]	Adolescent–parent dyads, N = 257	Parental monitoring, family conflict		Diabetes self-care
Cunningham et al. (2011) [15]	Adolescents and their caregivers, N = 147	Caregiver depression		Diabetes-specific burden

## Data Availability

Data can be made available upon reasonable request.

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
