# Peer review of "Exploring the Mediating Role of Parental Anxiety in the Link between Children’s Mental Health and Glycemic Control in Type 1 Diabetes"

_ijerph, 2023, doi:10.3390/ijerph20196849_

Round 1

Reviewer 1 Report

Dear Authors,

I would like to begin my comments by saying that this study has clear and opportune objectives, and I believe it is an interesting study. However, I find some comments or methodological decisions in it that are not clear, or even I think that some of them are wrong, and I would like to point them out for possible correction.

Most of these comments are in the Materials and Methods section, but there are others in the Introduction and in the Results. I will comment on each of them.

1. Introduction

On lines 45-48 says that "... as well as a three-way interaction among glycated hemoglobin (HbA1c) - a key criterion for diabetes compensation, frequency of blood glucose monitoring, and diabetes-related stress [11] ...". What does three-way interaction mean? It would be good to explain this sentence better.

Line 48 talks about "... self-control...", and it is not clear if self-control is the same as glycaemic control. Is it the same? Is it measured in the same way?

2.1 Participants

In lines 74-75 it says: "One of the child's parents, mostly mothers, participated in the survey ", and yet,

-       There are no results on the percentage of mothers and fathers.

-       No analysis has been done on the possible differences that may exist on parental characteristics when referring to mothers or fathers?

-       Is it possible that the results provided in the regression models may be different among adolescents when the parental factors refer to mothers or fathers?

In lines 81-82 and table 1 it is stated that the study population (or study group) is N=502. But actually the sample used in the regression model analyses has a size of N=251. The variables related to fathers or mothers are used as variables of the adolescents' environment. I think this is confusing in the text.

2.3 Statistical data analysis

The statistical analysis is confusing

-       In the first paragraph (lines 110-114) it is stated that stepwise regression models are performed to evaluate possible predictors of glycosylated hemoglobin level among parental factors. I assume that the regression models used are general linear models, statistical methods to relate some number of independent variables(IV) continuous and/or categorical variables to a single/multiple Dependent Variable (DV). Sometimes general linear models are confused with generalized linear models because in some manuals both are referred to as GLM. I think it would be good to clarify what type of models have been used in this first exploratory analysis.

-       Also, in the general linear models, assume a normal distribution for the response variable, but normality has not been assessed

-       On the other hand, at the end of the first paragraph (line 115) it is said that a number of associations and correlations are obtained. I do not understand what you mean. What associations and correlations are obtained? What for? Is’t not enough with the regression models analyzed?

-       In the next paragraph (lines 116-121) it says that the linear association between glycated hemoglobin (HbA1c) and GAD-7 and PHQ-9, ...is analyzed with Spearman's correlation coefficient because of non-normality.

-       I don't understand why normality is tested now and not before.

-       Which variables were non-normal?

-       There were no normality problems in the previous regression analysis?

-       Also, Spearman's correlation does not analyze the linear association of the variables.

-       I think it would be good to clarify the statistical analysis and the models that were used.

-       Regarding the third paragraph, it says that "Generalized linear models (GLM) mediation analysis was performed to determine the potential mediating effect on the relationship between dependent and independent variables"?

-       I think there is some confusion between general linear models and generalized linear models.

-       I guess it should say general linear models mediation analysis.

3. Results

As I said before, the regression models performed did not test the normality of the DV.

It is not clear which independent variables have been used in the regression analysis. Have all the variables related to parents and socio-economic factors listed in the Appendix table been included?

I do not understand what is the contribution of the paragraph on lines 148-154 on the correlations performed, once the first paragraph (lines 131-139) reports the best regression model obtained with the parental variables and the variables GAD-7 and PHQ-9 on the child's glycosylated hemoglobin.

Regarding GLM mediation analysis described in lines 157-170:

-       What about parental IV education, which was found to be significant in the initial regression model? Nothing is said in the mediation analysis. Is it possible to analyze a mediation model with the covariate parental education?

-       What is the meaning of the parameters Estimate and beta? Is Estimate the regression coefficient and beta the standardized coefficient? Please specify.

-       Figure 4 contains a value of the coefficient beta=0.113 (from child GAD-7 to Parent GAD-7) that does not correspond to table 3. It should be 0.669?

-       Finally, I think table 3 is confusing and should be reordered. In my opinion, it should first show the fits of the simple models titled "Component" first, and then of the indirect-direct-total or direct-indirect-total models.

Reviewer 2 Report

Thank you for allowing me to review this paper. I reviewed it with great interest, as it is current and vital. It deals with interesting topics, mostly how health disorders affects children's mental health (teenagers) . In that particular case reflections about parents anxiety and stress were taken into account.

Presented manuscript is well-written and well-organised, methods used in the study were well-described as well. The issue is adequately presented and authors have given the readers theoretical background for T1D.

Moreover, discussion is wide, including connections to relevant publications on the field. Only one suggestion to the authors, which could be helpful for the future readers. Please consider to develop last section of the manuscript (conclusions). It is too "general" and short in relation to other paragraphs. Perhaps relocate several thoughts from discussion is worth to mention there.

Reviewer 3 Report

Dear Authors and Editors,

I am very honoured to receive this article for reviewing.

It is very interesting and important research project presentation.

1. Introduction - might extended

2. the research aim and the conceptual model described in appropriate way.

3. material and method - the conceptual model includes socio-economical as well as parental factors however they are not testing in statistical analysis. There is the gap between theoretical and empirical concept - it must be improved

4. results and their discussion are presented preseted properly .

5. conclussions are valuable for practice and family therapy

6. References - Please take into concideration to add some other references, eg Fryt, J., Pilecka, W., Smolen, T. (2013) Importance of symptom control: self-regulation in children with diabetes type 1 and asthma, Studia Psychologiczne, 51 (3), 5-18.

Round 2

Reviewer 1 Report

Dear authors,

I am glad to have been given the opportunity to contribute to improving the manuscript.

I have reviewed your corrections, and I think it is acceptable for publication.

Kind regards